# More than 30 Years of PVC Recycling in Europe— A Critical Inventory

Uwe Lahl * and Barbara Zeschmar-Lahl 

BZL Kommunikation und Projektsteuerung GmbH, D-28876 Oyten, Germany; bzl@bzl-gmbh.de
* Correspondence: ul@bzl-gmbh.de

**Abstract:** PVC has a special status, as chlorine is a component of the polymer molecule. The properties of chlorine are the reason why the polymer molecule needs additivation. PVC is the mass plastic to which the most diverse and quantitatively largest number of additives are added. This makes PVC difficult to recycle. More than three decades ago, the PVC industry announced its commitment to improve the sustainability of the material through material recycling. We analysed the latest figures from the European PVC industry, ensuring that the statistics included the quantities that enter the market as recyclate. We also analysed the significance of replacing virgin PVC with recyclates. We conclude from this that, after a good three decades, the recycling result is rather meagre. The lion's share of PVC waste in Europe is still going to waste-to-energy plants, where it tends to be a nuisance. The many announcements to close the chlorine cycle via waste incineration have not got very far either. And the announcements to expand chemical recycling in parallel have not been successful. On the basis of this stocktaking, we have analysed, in a second separately published part, which conclusions can be drawn for regulatory measures, building on a current ECHA investigation report.

**Keywords:** PVC; material recycling; chemical recycling; chlorine cycle; additives



## 1. Introduction

The history of the chemical industry over the last 150 years includes a large number of pioneering developments such as the synthesis of dyes (fuchsine, alizarin, indigo), electrolysis (e.g., extraction of chlorine and caustic soda from water and common salt), the Haber–Bosch process (production of ammonia from atmospheric nitrogen), or the Fischer–Tropsch synthesis (production of hydrocarbons from carbon monoxide and hydrogen). However, it is also characterised by damage to humans and/or the environment, which in some cases has been recognised only many years after the actual event. This particularly applies to chlorine chemicals. In the early 1970s, for example, a chemical (DDT = Dichlorodiphenyltrichloroethane) was banned for the first time due to its recognised damage or risks (persistence, biomagnification, and toxicological effects). Following an environmental disaster (dioxins) caused in 1976 by an accident at a chemical plant in Seveso, northern Italy, Germany responded in 1980 with the Hazardous Incident Ordinance, followed, in the EU in 1982, by the so-called Seveso Directive [1]. In the following years, the production and use of certain persistent organic pollutants (POPs) have been banned worldwide since the Stockholm Convention came into force.

At that time, the risks of chlorine chemistry were the subject of intense debate. The plastic PVC (polyvinyl chloride) was also part of this debate, e.g., in Germany [2], Austria [3], or in the US and Canada [4]. Parallel to the debate about a ban at federal level, there were, in the 1980s, various municipal initiatives in Germany to stop PVC being used in construction projects because PVC was not considered sustainable. The main applications of PVC in the construction sector are pipes, window profiles, rigid films (=rigid PVC applications) and cables, floorings, and synthetic leather (=flexible applications). One of the first cities to ban PVC in construction projects was Bielefeld [5], followed by further cities

such as Dusseldorf, Nuremberg, Munich, and Vienna. The main arguments and concerns at the time were:

- In the event of fire, PVC serves as a precursor for the formation of toxic substances such as dioxins, and the fire residues are hazardous waste [6–8].
- PVC interferes with waste disposal (chlorine and toxic additives) [6].
- PVC is not sustainable (because of, e.g., problematic additives (heavy metals, phthalates), poor recyclability, mercury emissions from chlor-alkali electrolysis (amalgam process)) [7,9].

Many cities and municipalities subsequently issued a "PVC ban" for public buildings (i.e., products containing PVC were excluded from tenders for construction work). In Austria, for example, PVC-containing floor coverings, wall coverings, plastic pipes, and electrical installations are excluded from the procurement of materials for the new construction and renovation of service buildings (office buildings, educational buildings, sports and event venues, and healthcare buildings and hospitals) [10].

In order to counter the threat of government regulation of PVC, the manufacturing and processing industry in Germany has launched its own initiative. Behind the PVC industry's "Plastics Cycle", "PVC cycle guarantee" and "Global Recycling" concepts of 1988 and the AGPU (Arbeitsgemeinschaft PVC und Umwelt e.V., i.e., Working Group on PVC and Environment), which was founded for this purpose, lies the idea of recycling PVC waste and the chlorine it contains, and thus solving the respective waste and sustainability problem [11–13] (see also Section 2.4). Of course, an ideal circular economy entails much more than only recycling, i.e., a cascading of material and energy use, starting with reuse, refurbish, etc.

In 2000, the AgPU recycling promise was "europeanised", probably driven by the public consultation process initiated with the EU Commission's PVC Green Paper [14]. With "Vinyl 2010", the European PVC industry presented a ten-year program aimed at moving towards sustainability [15]. The sustainability pledge also included a commitment to promote recycling throughout Europe. The pledge was then given different names for the following decade: "VinylPlus", and most recently (2021) "VinylPlus 2030". We will stick with 'Vinyl' in the following sections.

This article takes a closer look at the PVC industry's recycling promises, critically examines the reported successes of material recycling, and analyses the possible causes for the achieved recycling rates. The status of chemical recycling to date is also discussed. The article ends with an outlook for the second part of our review about PVC.

## 2. Materials, Methods, Definitions, and Target

### 2.1. PVC Polymers and PVC Compounds

The basic building block of PVC is chloroethene ($H_2C=CHCl$), better known as vinyl chloride monomer or VCM. In the United States and Europe, VCM is mainly produced based on chlorination ($+Cl_2$) of ethylene followed by dehydrochlorination ($-HCl$) of the produced ethylene dichloride (EDC):

Ethylene ($H_2C=CH_2$) + Chlorine ($Cl_2$) $\rightarrow$ Ethylene dichloride/EDC ($H_2ClC–CClH_2$)

EDC ($H_2ClC–CClH_2$) $\rightarrow$ vinyl chloride monomer ($H_2C=CHCl$) + hydrogen chloride (HCl).

In industrialized countries, ethylene is mainly produced today by cracking natural gas, crude oil, ethane ($H_3C–CH_3$), or higher hydrocarbons. It is possible to convert ethane—which is cheaper than ethylene and nearly always available from natural gas—directly to vinyl chloride, e.g., by high-temperature chlorination using chlorine ($H_3C–CH_3 + 2Cl_2 \rightarrow H_2C=CHCl + 3HCl$), or by high-temperature oxychlorination using oxygen and hydrogen chloride instead of chlorine ($H_3C–CH_3 + O_2 + HCl \rightarrow H_2C=CHCl + 2H_2O$). The production of VCM from ethane has not yet become established. "Developing an ethane-based technology would be a breakthrough for VCM manufacturing in the future. However, despite its great potential, an ethane-based technology is still under exploration" [16].

In the next step, the vinyl chloride monomers are polymerized to polyvinyl chloride (PVC), see Figure 1. In sum, PVC consists of chlorine and hydrocarbons (57% resp. 43% by weight).

**Figure 1.** Scheme: Polymerization of vinyl chloride monomers (VCM) to polyvinyl chloride (PVC).

Currently, PVC is one of the most important polymers worldwide in terms of volume (see Table 1). Behind polyethylene (PE; here as the sum of HDPE and LDPE, LLDPE) and polypropylene (PP), PVC is in third place with a global production volume of a good 51 million tonnes per year. Global PVC production is forecast to increase to almost 60 million tonnes per year after 2030 [17].

**Table 1.** Global polymer production 1990–2019, in million tonnes per year.

| Polymer | 1990 | 2000 | 2010 | 2015 | 2016 | 2017 | 2018 | 2019 |
|---|---|---|---|---|---|---|---|---|
| PE (sum) | 30.0 | 54.6 | 81.9 | 98.2 | 100.1 | 103.2 | 106.5 | 109.8 |
| *HDPE* | *14.4* | *26.6* | *40.7* | *49.8* | *50.8* | *52.3* | *53.9* | *55.5* |
| *LDPE, LLDPE* | *15.7* | *27.9* | *41.2* | *48.4* | *49.3* | *50.9* | *52.6* | *54.3* |
| PP | 20.8 | 37.2 | 55.4 | 65.4 | 66.6 | 68.5 | 70.7 | 72.8 |
| PVC | 14.7 | 26.2 | 38.6 | 45.3 | 46.1 | 47.8 | 49.8 | 51.4 |
| PS | 6.4 | 11.2 | 16.3 | 18.9 | 19.2 | 19.8 | 20.5 | 21.1 |
| PET | 6.6 | 12.2 | 18.5 | 22.5 | 23.0 | 23.6 | 24.2 | 24.9 |
| PUR | 5.4 | 9.5 | 13.9 | 16.1 | 16.4 | 16.9 | 17.5 | 18.0 |
| Total | 129.9 | 234.0 | 348.9 | 412.8 | 420.0 | 432.2 | 446.2 | 459.7 |

Source: Global Plastics Outlook—plastics use by polymer [18].

As the data in Table 1 only relate to pure polymers, the volume of plastic products made from PVC is likely to be higher than the data stated. The PVC polymer is mixed (compounded) with additives for the respective area of application. As pure PVC is hard, becomes brittle at low temperatures, and begins to decompose at temperatures above 160 °C, which impairs processability (extrusion or injection moulding), various additives are added to the polymer to eliminate these weaknesses, such as plasticizers, stabilizers, and other additives. The weight proportion of additives in the compound is between four and just under 20% by weight for rigid PVC products (PVC-U, U = unplasticized), and up to 50% by weight for soft PVC products (PVC-P, P = plasticized) [19]. Table 2 gives a survey on applications of PVC and the typical composition of PVC compounds.

The sales volume of uncompounded PVC in the EU amounted to 5.2 million tonnes in 2021 (Eurostat, 2023b, cited in [19]), which (according to the European Chemicals Agency ECHA) corresponds to around 6.8 million tonnes of PVC compounds. In addition, 0.5 million tonnes of unmixed PVC are imported into the EU each year, and 1.2 million tonnes are exported to countries outside the EU (Eurostat 2023b, cited in [19]). Adjusted for these import and export quantities, 4.5 million tonnes of unmixed PVC-polymers were therefore used in Europe, corresponding to 5.9 million tonnes of PVC compounds. Almost

70% of this was used in the construction sector, mainly for pipes, floor coverings, cables, and window/door frames [21]. A good 50 to 85% of this quantity is used in rigid, and 35 to just under 50% in flexible PVC applications (based on data in [19]).

**Table 2.** Applications of PVC and typical composition of PVC compounds (in weight-%) [20].

| Application | PVC Polymer | Plasticizer | Stabilizer | Filler | Others |
|---|---|---|---|---|---|
| Rigid PVC applications (PVC-U) | | | | | |
| Pipes | 98 | - | 1–2 | - | - |
| Window profiles (lead stabilized) | 85 | - | 3 | 4 | 8 |
| Other profiles | 90 | - | 3 | 6 | 1 |
| Rigid films | 95 | - | - | - | 5 (1) |
| Flexible PVC applications (PVC-P) | | | | | |
| Cable insulation | 42 | 23 | 2 | 33 | - |
| Flooring (calendar) | 42 | 15 | 2 | 41 | - |
| Flooring (paste, upper layer) | 65 | 32 | 1 | - | 2 |
| Flooring (paste, inside material) | 35 | 25 | 1 | 40 | - |
| Synthetic leather | 53 | 40 | 1 | 5 | 1 |

(1) incl. approx. 0.5% stabilizer.

### 2.2. Methods for Recording the Volumes of Recycled Material

As mentioned above, to become sustainable, the PVC industry promised recycling. Figure 2 shows the calculation points used for the statistical records of material recycling. For lightweight packaging, the calculation point (1) was used for many years in Europe. This resulted in the known high recycling rates, but failed to record all recycling losses along the processing chain. Commission Implementing Decision (EU) 2019/665 from 17 April 2019 [22] prescribed calculation point (2) for statistics from 2020 onwards.

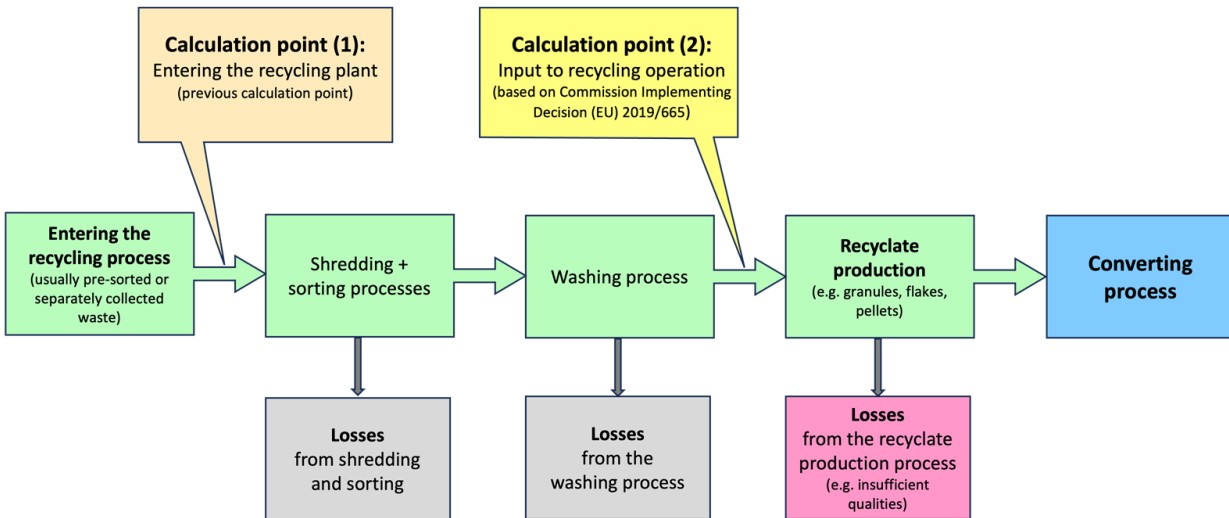

**Figure 2.** Calculation points for recording recycling quantities in the recycling process when recovering packaging waste (based on Conversio [23], modified).

However, in our opinion, calculation point (3) in Figure 3 would be more suitable because it also takes into account the losses during the production of recyclates, and thus statistically reflects the actual quantities of recyclates produced at the end of the chain. But calculation point (3) will not display the quality of recycling (share of downcycling). The central ecological reason for propagating material recycling is the replacement of virgin

material (in this case, virgin PVC). This would correspond to calculation point (4), which we consider to display the share of closed loop recycling, what would have, in general, better ecological results.

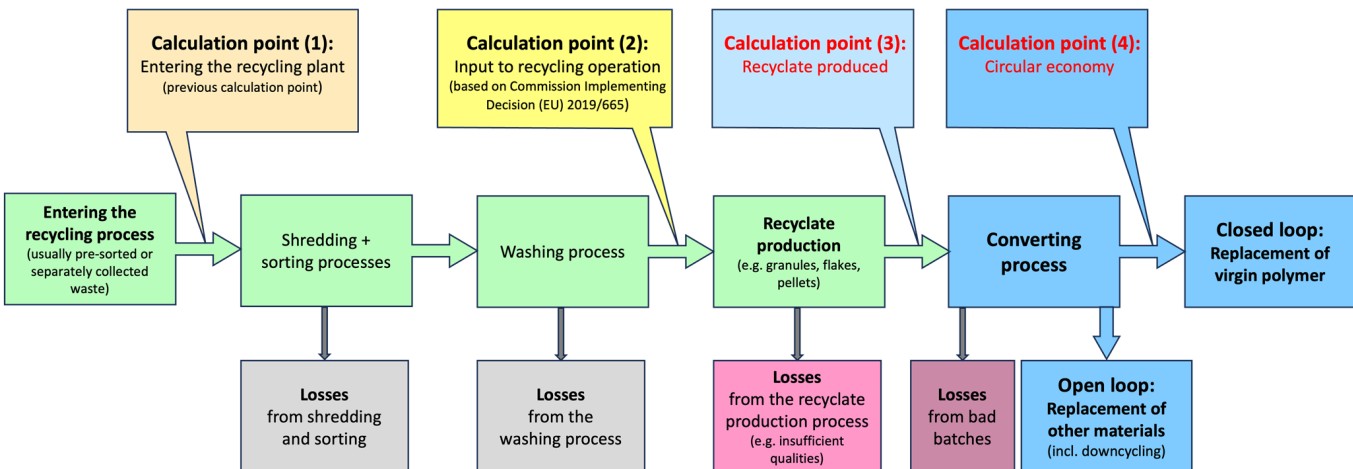

**Figure 3.** Calculation point for recording "real" quantities in the recycling process of PVC.

As part of the auditing of recycling data in Europe, the converter output has recently also been recorded, which would come closer to calculation point (4), as this would exclude unusable recyclates, for example [24,25]. However, calculation point (4) would also subtract products that do not contribute to closing the PVC loop (Figure 3: Open Loop box: PVC recyclates replace wood or concrete, for example). In the final state, the circular economy means that there is no longer any need for new material (in this case, virgin PVC) or that only the unavoidable loss of material is replaced by virgin PVC. On the way there, the amount of virgin PVC used must gradually decrease each year. The volume of the virgin market would have to become smaller and smaller. The extent to which this takes place is methodically recorded via calculation point (4).

### 2.3. Definitions: Recycling

There are different definitions for the recycling of plastics, particularly with regard to the distinction between chemical recycling and feedstock recycling/recovery [26,27]. In this article, we use the following definitions:

- Physical recycling, which includes:
  - Mechanical recycling: processing without dismantling the compound; the bond of PVC polymer and additive is retained; and the output are flakes or granulate, which are converted to new products.
  - Solvent-based recycling (purification, dissolution): use of chemicals (e.g., organic solvents) to dissolve the compounded plastic, whereby the polymer chain remains intact and can be reused for the fabrication of new products.
- Chemical recycling:
  - Depolymerization: This includes, e.g., thermal depolymerisation, chemolysis, and solvolysis. Here, the plastics are broken down into their building blocks—oligomers (partial depolymerization) and monomers (full depolymerization).
  - Thermolysis: Decomposition of the polymers by thermal processes (e.g., pyrolysis, gasification); the resulting fragments (monomers, hydrocarbons, CO, etc.) can be used as feedstock for the synthesis of other substances or as a raw material in other processes. Following this logic, waste-to-energy (WtE) with carbon capture and utilization (CCU) in the chemical industry is also part of chemical recycling.

- Energy recovery (thermal recycling): Use of the energy content of the plastic in incineration plants (waste-to-energy), cement plants, or power plants fired with solid recovered fuels (SRF).

Within circular economy thinking, material recycling is often categorized based on the product which is manufactured from the secondary raw materials [28]:

- "Closed loop is a recycling process whereby the recycled material is reused for the same market application as that of its previous life cycle (system-wide concept)".
- "Open loop is a recycling process whereby the recycled material is used for a different market application than that of the previous life cycle (system-wide concept)" [29].

Open loop recycling is often equated with downcycling. There is also no legal or official definition of "downcycling". Experts define downcycling as "a recycling process whereby the recycled material is used for a lower-quality market application than that of the previous life cycle, normally defined by a lower market value, as opposite to upcycling (system-wide concept), defined for plastics as: "the use of plastic waste, post-industrial or postconsumer, as a feedstock for the synthesis of value-added products, being polymers, molecules, or materials" ([30], cited by [29]). Other experts propose, based on a stakeholder process [31], that: "Downcycling is the phenomenon of quality reduction of materials reprocessed from waste relative to their original quality, where waste means any substance or object which the holder discards or intends or is required to discard. Downcycled materials count as recycled materials....".

Open loop recycling does not necessarily mean downcycling. "In fact, there might be cases where open-loop recycling produce recyclates of adequate quality for another application, rather than unnecessarily pushing closed-loop recycling processes to increase the technical properties of recyclates to meet the standards of their virgin counterpart. Open-loop recycling can still contribute to meet demand of the virgin resource while incurring environmental benefits as long as there is an unsaturated market application for that secondary material which is ready to absorb it (...). However, one should also bear in mind that these substitutions would not occur when such outlet markets suddenly become saturated (e.g., because of closed-loop recycling initiatives taking place within that sector or bans in export)" [29].

If recycled PVC (rPVC) replaces virgin plastic 1:1, this is undoubtedly recycling. However, the technically possible recyclate admixture is limited: pipes 25%, cable sheathing 30%, floor coverings 30%, window profiles 60%, and other profiles 25% (AgPU, cited in [32]). In most cases, a formerly high-quality primary product becomes an intermediate layer, sub-layer, or a low-grade product. "Even if downcycling is not entirely avoidable, we can still foster high-quality recycling and implement a cascade of slowly and gradually downcycled materials" [31]. But downcycling entails the risk that it will only have minor environmental benefits. Mixing without any relevant contribution to the product properties is not recycling, it is dumping in products.

The circular economy means that we need to organize more than just a material cycle. It is true that this issue is very forward-looking in the construction sector. However, it must be stated that the compounds of virgin PVC and recyclates will move further down the cascade of downcycling in the next future cycle.

### 2.4. Target: Sustainability through Recycling

2.4.1. Germany (1988)

Behind the sustainability promise of the AgPU (Working Group on PVC and Environment) in 1988 was the plan to recycle PVC waste as far as possible to substitute virgin PVC and thus eliminate the waste problem [13]. The plan was based on an industry concept, which ideally meant, for example, that a new window frame would be made from a used PVC window frame. Virgin PVC would not be essential any more. Old PVC goods (post-consumer) therefore had to be collected and recycled in a sector-specific manner. AgPR (Arbeitsgemeinschaft PVC-Bodenbelag Recycling, https://agpr.de/en/,

accessed on 31 March 2024) was founded for the recycling of floor coverings, AfDR (Arbeitsgemeinschaft für PVC-Dachbahnen-Recycling [33]) for roofing membranes, the Wavin subsidiary Replast for pipes [34], and the companies Fenster Recycling Initiative (FREI, now: REWINDO www.rewindo.de, accessed on 31 March 2024) and VEKA-Umwelttechnik (https://www.veka.com/divisions-brands/veka-recycling, accessed on 31 March 2024) for window and other profiles. As a rule, the leading producers and processors were involved in these projects as shareholders [35].

This approach naturally meant that the respective product manufacturers could be held more accountable. However, it also meant that chemically similar waste (e.g., window profiles) was collected separately, which ultimately promised higher-quality recycling. In 1997, the situation was as follows [36]:

- The manufacturers of PVC flooring had built a recycling plant for around 2.6 million Deutsche Mark (DM) (about 1.3 million EUR), but only 10% of the plant was utilised with post-consumer PVC flooring waste. The recycled material was used as a back coating.
- The manufacturers of roofing sheets had built a recycling plant for a good 2 million DM (about 1 million EUR). The plant was 20% utilised with PVC roofing membranes. The recycled material was used as a centre layer in new products [37–39].
- Only just under 5% post-consumer PVC was used in the pipe recycling plant (mainly PE was recycled). Low-quality products such as support pipes or winding cores for carpeting were produced.
- The window manufacturers had set up two recycling plants. One was built in Behringen for just under 18 million DM (about 9.2 million EUR). The plant was well utilised, but mainly processed pre-consumer offcuts. Only just under 5% of post-consumer PVC was recycled at the time. The second plant in Rahden processed off-cuts and post-consumer PVC separately and had a slightly better capacity utilization of 30%.

For the question of whether this has been recycling or downcycling in individual cases, see Section 2.3. Overall, the balance at that time was rather poor [40,41]. So, the question arises: How has the recycling of PVC compounds developed since then?

### 2.4.2. Europe (Since 2000)

In 2018, 'Vinyl' has committed to recycle 900,000 tonnes of PVC by 2025 and 1 million tonnes by 2030 as part of the European Commission's Circular Plastics Alliance (CPA) [42]. The Alliance endorses the ambitious target that by 2025 at least 10 million tonnes of recycled plastics should find their way into products and packaging in Europe each year (hereafter referred to as "the 10 million tonnes target"), helping to deliver the circular economy with a life cycle approach. 'Vinyl' points out: "Over the last two decades, reporting annually and overseen by an independent Monitoring Committee, Vinyl 2010 and VinylPlus succeeded in meeting their targets" [15]. Figure 4 shows the current recycling balance—a considerable increase.

'Vinyl' states that 813,266 tonnes of PVC waste have been recycled in 2022 ("PVC recycled within the VinylPlus framework"). "Taking into account also PVC waste streams that do not have formally established collection schemes (. . .) and rPVC volumes not yet traced and certified through the Recovinyl® system, Conversio data estimates that 35% of PVC waste is currently being recycled in Europe [43]".

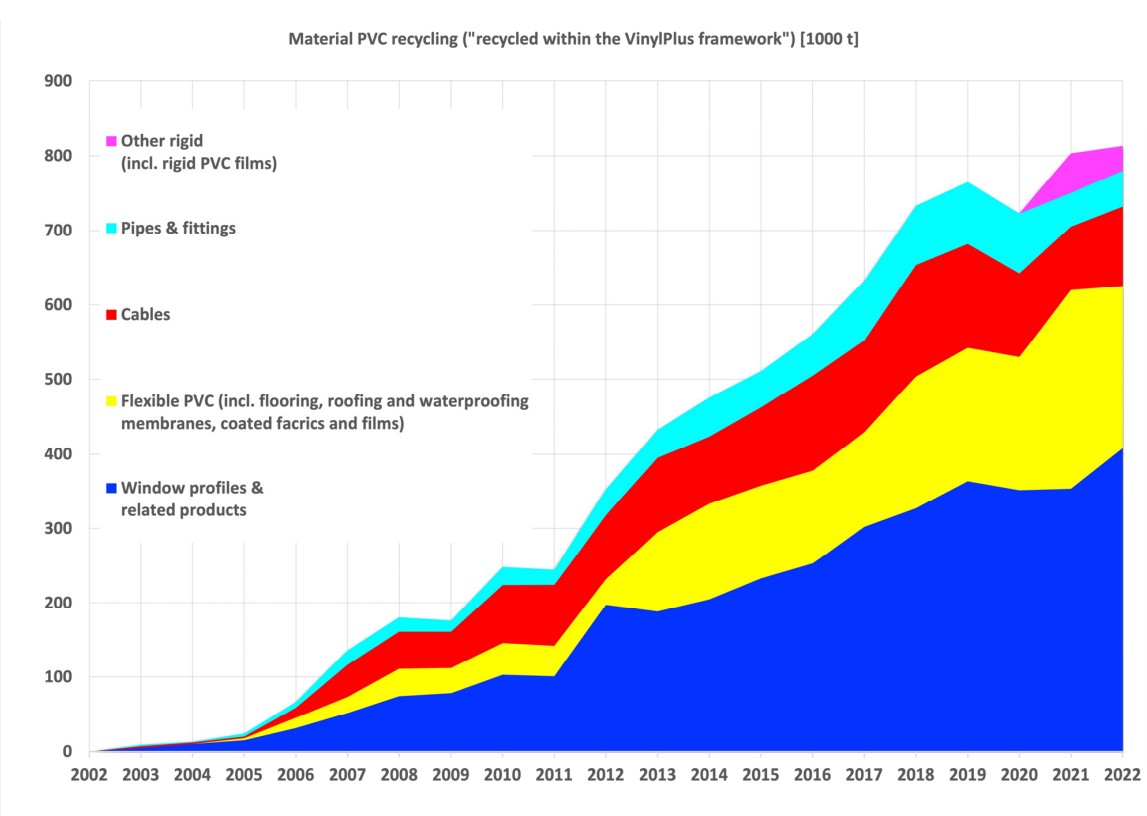

**Figure 4.** Material recycling of PVC ("recycled within the VinylPlus framework") 2002 to 2022 (based on data published by VinylPlus [44]).

### 3. Results: 35% Recycling Rate in Europe—An Analysis

Although a material recycling rate of 35% is not quite at the target of a closed loop, it would still be progress, and it is for the entire European Union. But is this figure reliable?

#### 3.1. Reference Value

The layers in Figure 4 contain both pre- and post-consumer recyclates. However, if the recycling volumes from post-consumer material (305,594 tonnes ([43], p. 44)) are related to the total volume of post-consumer PVC waste (around 2.5 million tonnes of PVC waste), the recycling rate for 2022 is 12% (also according to the ECHA ([21], p. 60)). 'Vinyl' states [45]: "PVC post-consumer recycling rate was not 12% but 24% according to Conversio 2021. This is a factual error from the ECHA report. See . . . [46]. The amount quoted by ECHA corresponds to the amounts that have been certified in the Recovinyl system, i.e., 290 kT, which does not catch all post-consumer recycling. However, a Conversio study from 2020 estimates that 599 kT of post-consumer waste is recycled".

The Conversio study of 2021 has not been published so far. Therefore, in this article, we will stick to use the certified recycling data of 'Vinyl', respectively, 'Recovinyl', also because these data were not extrapolated.

Looking back, it can be seen that a good 260,000 tonnes of post-consumer waste were already recycled in 2010 [47,48]. In 2021, the figure was a good 290,000 tonnes [49]. The increase in Figure 4 was therefore primarily achieved by increasing the collection of pre-consumer recycling volumes. For post-consumer waste, there was only a slight increase (from 260,000 to 305,000 tonnes, i.e., by 45,000 tonnes, which corresponds to around 17%) in this period. 'Vinyl' points out: "If one considers post-consumer recycling only the increase is actually quite dramatic from 2000 until 2020: from 100 kT to 599 kT, this is a 9.3% growth rate, much more than 'virgin resin' [45]". However, this statement shows two problems: 1. With a low initial value, even a small upward changes lead to high percentage increases. 2. These new Vinyl figures for post-consumer waste recycling cannot be reconciled with

the previous figures for Vinyl that we have reproduced in Figure 4. These new figures are significantly higher than what Figure 4 contains. The data in Figure 4 is certified according to Vinyl. The new figures are based on extrapolations of data from surveys of recycling companies and extrapolations made using the expertise of plastics associations. They have not been certified, and are therefore not reliable in our view. We will therefore not base our analysis on these new figures, but we will not conceal them either.

'Vinyl' confirms that the increase of post-consumer recycling in the second decade of the Voluntary Commitment has slowed down. "Several factors explain this:

(1) The new VinylPlus 2030 commitment focused on all waste streams including pre-consumer as it was acknowledged that the full recycling potential had not yet been achieved there in part because of the strong focus of previous VinylPlus commitments on post-consumer waste.
(2) New restrictions have been put in place decreasing the number of potential outlets for recyclates and limiting the ability to recycle certain products (e.g., decrease in recycling in sheets following the restriction of DEHP (Di(2-ethylhexyl) phthalate), lead in PVC restriction). This is probably a trend that will be reinforced short term until new projects enable to unblock recycling potential (e.g., sorting of legacy additives, chemical recycling, dissolution technologies. . .).

The assessment of how to tackle those challenges is part of the foreseen VinylPlus 2030 commitment mid-term target review (in 2025)" [45].

Recycling of pre-consumer waste is laudable and has always been practised, where economically advantageous. However, contributions from post-consumer recycling are more crucial for closing material cycles. This is another reason why the current voluntary commitment of the plastics industry within the Circular Plastics Alliance (CPA) (see Section 2.4.2) [50] only includes material from post-consumer recycling [25]. However, the PVC industry and 'Vinyl' have been able to obtain special treatment from the Commission; they are allowed to include the quantities from pre-consumer recycling in their voluntary commitment [25]. The argument in favour of this special treatment is that, for PVC, better recyclate qualities can be achieved with the mixture of pre- and post-consumer waste [51].

Figure 5 based on data from 'Vinyl' [43] shows the volumes of PVC compounds recycled in the EU in 2022 (813,266 tonnes in total), broken down by product and origin and separated for pre- and post-consumer waste. It is clear that post-consumer recycling results are mainly achieved through the recycling of window frames (56%). This has already been the flagship in 1997 [36,40].

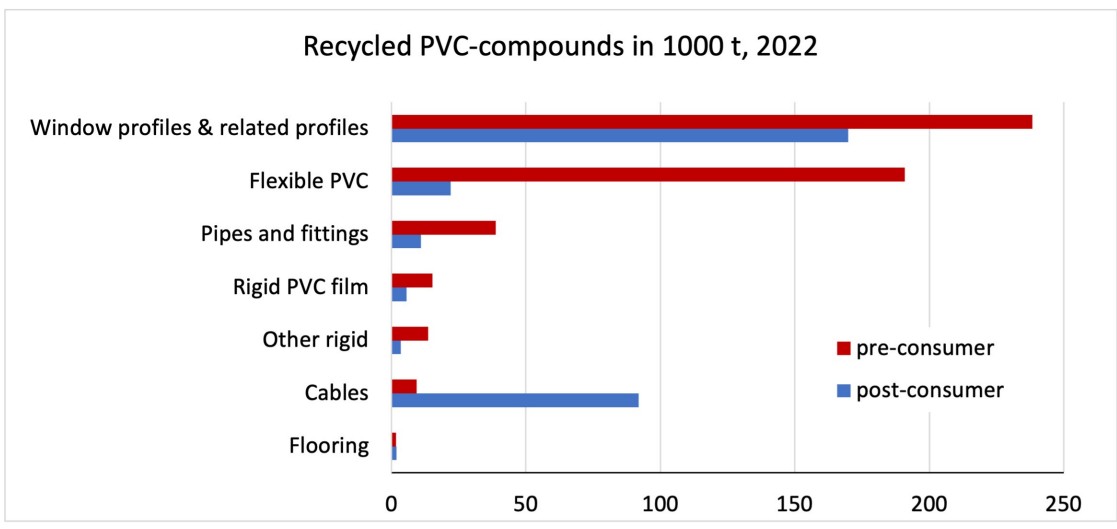

**Figure 5.** PVC compounds recycled in the EU in 2022 (813,266 tonnes) (based on data according to VinylPlus [43]).

For recycling of PVC cables, ECHA sees a need for clarification [19]: "This rate seems to contradict the information received from plastic recyclers which suggests that PVC in electrical and electronic equipment mostly ends up in the heavy fraction of the shredder light fraction which is subsequently incinerated". Following 'Vinyl' [45], PVC cable recycling is well established and "verified as part of the Recovinyl network. Cable recyclers are mostly not members of Plastics Recyclers Europe". We are not convinced by this explanation. For this, we also see the need for clarification.

### 3.2. Recycling or Downcycling

Let us analyse the bars in Figure 5 in more detail. What method respective to the calculation point is used for the statistic of post-consumer recycling? Is there, following calculation point 4 in Figure 3, a replacement of virgin PVC?

For plastics such as PVC, this poses a particular difficulty in the assessment, because recycling of post-consumer waste—as experience from the 1990s has already shown—was, in our opinion, generally downcycling, even if collected separately (see Section 2.3). Overall, we have not been able to find reliable data on the extent recyclates are produced out of post-consumer PVC (calculation point 3) or to which extent PVC recyclates replace virgin PVC (calculation point 4). In our opinion, this is where the greatest need for action lies in order to achieve greater data transparency.

In terms of the life cycle assessment (LCA), material recycling in general is only superior to waste-to-energy options if credits for the substitution of virgin plastic can be taken into account. If recyclates replace lower-quality materials (downcycling, see Section 2.3), then the eco-balance credits are rather low. This must be checked in each individual case. The question is at what stage of the "downcycling cascade" these new products are. For recycling, 'Vinyl' gave the following definition: "Recycled PVC is a discarded PVC product, or semi-finished product, that is diverted from waste for use within a new product; processing waste is included provided it cannot be reused in the same process that generated the waste" [52]. The term "use within a new product" is of importance, as it is not identical with substitution of "virgin" PVC (→open instead of closed loop). 'Vinyl' states that "one of the main issues of recycled product is their colour (the mix will often be greyish), not making it suitable for certain application. The solution is to find new applications where colour is not fundamental, or to use the recyclate in inner layers (co-extrusion)" [45].

For 2022, 'Vinyl' gives the following data: 562,000 tonnes recyclate uptake at plastic converters (corresponds to calculation point 3 in Figure 3) was registered in the Recovinyl system. However, it should be noted with regard to this figure that the survey both relied on recyclers output and the new registration system for recyclate uptake at plastics converters. 'Vinyl' points out that "the picture is still incomplete as this is a new registration system, and many convertors have not yet joined (the number of converters is bigger than the number of recyclers)".

The end-application of converting output as products was almost 468,000 tonnes (=83%) [45]. The difference of about 94,000 tonnes (17%) went elsewhere, presumably to energy recovery. The end-applications were:

- Windows and profiles: ...................................39%
- Floor coverings: ............................................25%
- Traffic management: ......................................18%
- Pipes: .............................................................10%
- Building & construction—other: ...................6%
- Horticultural and stable equipment: .............1%
- Coils and mandrels: .......................................0.18%
- Packaging: ......................................................0.16%

'Vinyl' points out, that "traffic management covers a range of products and should not be understood as downcycling. There are indeed some bases for holding fences in

replacement of concrete, but this is a minority. Typical applications would be traffic cones, cable bridges, and structures for drainage" [45]. We follow this argument partly.

Figure 5 and the data provided by 'Vinyl' show that window profiles (and doors, etc.) are by far the most important contributors to post-consumer recycling. Historically, this certainly also has to do with the fact that this waste is very easy to recognize and to grab from construction waste. Today, post-consumer material may only be used for the internal profiles due to the contamination with meanwhile banned hazardous substances and regulatory requirements. The contact surfaces on the outside must be made from virgin PVC as part of a co-extrusion process. E.g., at REHAU, in the last decade, about 15% of the tonnage required for the inner core of PVC windows consisted of recyclate [53]. In 2020, new window products from REHAU already had a recycling share of 54%. By 2025, this proportion is set to increase to 65% [54].

The situation will become even more complicated in the future. Glass fibre reinforced window profiles (GRP) have been increasingly available for a good five years. This provides greater rigidity so that triple glazing can also be manufactured from plastic. Thermal insulation is also improved because the thermally conductive steel core of the frame can be dispensed with. However, there are very different GRP techniques on the market (connective sleeves with glass fibres, pultruded profiles, plastics with glass fibres in the matrix, etc.). There are also market participants that use other GRP-reinforced plastics, such as polybutylene terephthalate, which would further complicate the situation if market growth were to occur here.

These windows are expected to be increasingly found in waste from 2030 and beyond. It will then be necessary to separate the GRP-PVC material from the shredded and ground other PVC [53]. According to a market participant, this requires the GRP-PVC in the window frame to be provided with a fluorescent marker so that this PVC can be detected afterwards. For this, the technical infrastructure has to be established throughout Europe. Another market participant is sceptical here and considers PVC windows with GRP components to be only theoretically, but not practically recyclable [55]. For his own recycling practice, GRP windows are regarded as "impurities". As long as there is no consensus in the industry on the GFR complex, future material recycling (replacement of virgin PVC) is unclear.

The example of the dominance of window recycling also indirectly shows the difficulties of further expanding post-consumer PVC recycling. While windows, as the proverbial low-hanging fruit, are easy to separate from bulky and construction waste, this is much more difficult for the smaller types of waste. In addition, window recycling is again dominated by German companies. The market here is very centralized, which has made it easier for the players to reach an understanding. It is questionable whether this can be achieved for the other areas of PVC use in the construction sector, which tend to be dominated by SMEs, and whether companies can be won over on a voluntary basis for the recycling of materials whose formulation is unknown and where the former manufacturers may no longer even exist. This is also because the strategy to "hide" the recyclate inside the window frames (or pipes) is more difficult for other applications. We therefore expect post-consumer recycling in the EU to stagnate, perhaps even decline, over the next few years.

### 3.3. Recycling Rates in Germany

As shown above, PVC recycling has been practised in Germany for some time. Therefore, higher recycling rates should be expected here than for Europe as a whole. In fact, more detailed analyses show that the high recycling rates acclaimed by 'Vinyl' are achieved in Germany in particular. The most recent publicly available data for Germany is from 2021 and was collected by Conversio. The full report has not been published, and only an abridged version is publicly available [23]. Conversio shows figures for post-consumer recycling (previous calculation point (1): input into the recycling process), according to which, the recycling rate for material recycling of PVC from post-consumer origin in Germany in 2021 was 29% (189,000 of 658,000 tonnes). In relation to all PVC waste in total (the

sum of post-consumer and pre-consumer waste) and calculation point (1), the share of PVC waste going to energy recovery in 2021 was around 57%; 42% of PVC waste was directed to material recycling (including a very small proportion chemically recycled). Around 7000 tons of PVC waste (1%) was disposed of in landfills.

With its 658,000 tonnes, Germany accounted for the lion's share (81%) of the total of PVC compounds recycled within the VinylPlus framework in 2021 (about 810,000 tonnes). Regarding solely post-consumer PVC waste, Germany's share was 64% (189,000 of 295,273 tonnes).

In relation to calculation point (2), the breakdown shifts to 62% energy recovery and 38% material recovery. A good two thirds (27,000 tonnes) of the losses from the recycling process (Figure 3, pink box), totalling 40,000 tonnes, were used as substitute fuel (cement plants, power plants, etc.), and one third was used for energy recovery in waste incineration plants. These 40,000 tonnes correspond to just under 5% of the total PVC waste mass collected in 2021 (861,000 tonnes). The material recycling of post-consumer PVC according to calculation point (2) in Germany is then 23% (see Section 3.1: Conversio data are not certified).

We have not been able to obtain data on recycling calculated according to calculation point 3 or 4. It can therefore only be assumed that these data will be below 23%.

The PVC industry and 'Vinyl' certainly do not share our low statistics (23% for Germany vs. 24% (Conversio) to date in Europe, see Section 3.1) and our pessimistic assessment of future PVC material recycling. For this reason, we suggest that industry's current voluntary commitment to the EU Commission could be conceptually developed further after 2025 to provide for a gradual replacement of virgin PVC with recyclates over several years. The cycle should then be closed by 2040 at the latest, except for the unavoidable loss of material. If one were to follow the general climate protection targets, a target for the closed loop would be 2040. We consider it highly questionable whether such a target can be realized for PVC, a plastic that is difficult to recycle. But it would be interesting to discuss this issue with the industry.

### 3.4. VINYLOOP®—A Review

Recycling of PVC by using solvents—VINYLOOP®—is often attributed to chemical recycling, but is a solvent-based (=physical) recycling process (see Section 2.3). It was first tested on a semi-industrial scale in a plant in northern Italy in 2002 [56]. In this process, old post-consumer PVC products, such as cable sheathing, were dissolved in a solvent, and the PVC contained was then precipitated out again. The advantage of the process was that PVC composites with other materials could also be processed. However, the meanwhile banned phthalate plasticisers contained, such as DEHP, proved to be an unsolvable problem. According to REACH, these plasticisers could no longer be placed on the market in the EU as of 21 February 2015. The Commission had granted a derogation with a limit value of 20% DEHP in the recyclate. However, due to the high concentration of phthalates in the soft PVC compounds, it was not possible to produce a recyclate that complied with this limit [57]. This is the main reason why the plant in question was closed in 2018 [58]. 'Vinyl', on the other hand, emphasizes that "the closure of Vinyloop was not due to the fact that the DEHP limit was not met. Actually, DEHP levels were considerably reduced, and a review report by ECHA allowed the process to continue with a reduced limit of 5% [59]. The decision to stop activities was related to financial reasons and a smaller customer base. It remains that the removal of those additives should be considered in the development of new solvolysis processes" [45].

### 3.5. Chemical Recycling—A Review

3.5.1. Recycling Options

The PVC industry's narrative has always included "feedstock recycling" or chemical recycling, which would open up completely new horizons. 'Vinyl' points out [45] that "in the case of PVC, both parts of the PVC molecule can be chemically recycled: the

hydrocarbon part (43%) can be recycled by processes suitable for hydrocarbon polymers, such as pyrolysis or gasification [60], while the chlorine part (57%) can be recycled and used in the production of rPVC as well as other raw materials and substances" (Vinyl cites Z. Hruska, Public Affairs Director at VinylPlus®): Chemical Recycling of Specific Plastics Waste Streams—Case PVC, in Conference Proceedings of the 10th International Symposium on Feedstock Recycling of Polymeric Materials, Budapest, 2019).

It is undoubtable that these recycling options exist. But theory is one thing, practice is another. The question is what really works in practice on an industrial scale and can therefore make a significant contribution to PVC recycling.

### 3.5.2. Hydrocarbon Recycling

'Vinyl' points out [44]: "There are different chemical recycling technologies suitable for plastics waste containing hydrocarbon polymers, such as pyrolysis, gasification, hydro-cracking and depolymerization. If the plastics waste streams contain heteroatoms such as, e.g., halogens, then these must be removed either before (via pre-treatment) or directly during chemical recycling process. . . . Some pyrolysis processes can tolerate high concentration of chlorine in the feedstock. For example, ARCUS Greencycling Technologies GmbH provides a chemical recycling solution for otherwise non-recyclable mixed plastic waste [61] streams that have undergone minimal pre-sorting or pre-cleaning. Their process is claimed to successfully handle a wide range of polymers, from PP, PE and PS to difficult-to-process PVC, ABS or PET".

Thermal processes (pyrolysis or gasification) could theoretically overcome the problem of legacy additives, because the hazardous additives are either broken down (organic substances) or eliminated for example via ashes (heavy metals). Small amounts of PVC within a plastic mixture will only result in operational challenges in thermal processes. For pure PVC or waste with high share of PVC, special technologies are needed. One of the most promising is de-hydrochlorination, which has the concept of "evaporating" the chlorine in the form of HCl from the polymer matrix. There exist different options when increasing the temperature above 300 °C.

The history of PVC recycling by 'Vinyl' is linked to many pilot projects for chemical recycling, all of which were put on display at their time: the Rotary kiln project with off-gas scrubbing (DOW and later SUEZ in Schkopau (GE)), pyrolysis (NKT-Watech (DK)), de-hydrochlorination (REDOP (NL)), the Stigsnaes process (DK), the Alzchem project (GE), which eliminates chlorine with a heated extruder before entering the process, or the KU Leuven project (BE) with the de-hydrochlorination in a ionic liquid media [52,62–66]. None of these projects was scaled up to an industrial plant. To date, not a single plant exists that can chemically recycle relevant quantities of PVC waste. The pyrolysis plant of ARCUS mentioned above (having started operation in 2022) is a commercial-scale pilot plant with a capacity of 4000 t/a (output: 2500 t/a), no large-scale industrial plant [67]. We will see what experiences ARCUS will gain concerning PVC.

### 3.5.3. Chlorine Recycling

The field of HCl recovery from waste incineration has been similarly unsuccessful to date. It was already clear to the experts in the 1990s: hydrochloric acid recovery from waste incineration plants is—from an economic point of view—the best way to recycle the chlorine content of PVC, as hydrogen chloride from flue gas scrubbing does not have to be neutralized, but can be returned to the economic cycle as hydrochloric acid (HCl) [7]. Similar to chemical recycling, a lot has been tried out here and put on display (MVR Rugenberger Damm, Halosep, The SOLVAir® Solution, etc.), but to date, nearly nothing is running in Europe in the operation of a waste incineration plant (for Switzerland, see below).

New ideas from 'Vinyl' take up the chlorine cycle again [68]: "In the VinylPlus® RecoChlor chemical recycling programme, the selected PVC wastes are thermally decomposed in modern waste-to-energy plants, which enables to recover chlorine either in the form of sodium chloride (RecoSalt, dry process) or as diluted hydrochloric acid (RecoAcid, wet

process). ... Both RecoSalt and RecoAcid processes lead at the end to the production of new chemicals and/or metals which can be sold on the market". The polymeric (hydrocarbon) part, on the other hand, is used for energy recovery.

In Switzerland, metal recovery from filter ash produced during the treatment of municipal waste is obligatory from 1 January 2026 [69]. In 2018, already "more than 60% of fly ashes in Switzerland were treated according to the FLUWA process, which represents a state-of-the-art technology" [70]. In 'Vinyl's RecoAcid project at the waste incineration plant in Thun, HCl is separated from the flue gas and used to treat the flue gas dust (acid fly ash washing). This allows for heavy metals to be extracted from the fly ash as chlorides and recycled. The trial (that has meanwhile ended) has confirmed that wet flue gas scrubbing in municipal solid waste incineration (MSWI) plants "is a valid option for the recycling of PVC wastes which cannot be mechanically recycled" [43]. But this recommendation only applies to plants with wet flue gas cleaning (FGC). In Europe, however, many MSWI plants are equipped with a dry FGC system. A report for 2011 on 455 waste-to-energy (WtE) plants in 18 European countries (Austria, Belgium, Czech Republic, Denmark, Finland, France, Germany, Hungary, Ireland, Italy, Netherlands, Norway, Portugal, Slovakia, Spain, Sweden, Switzerland, and the United Kingdom) showed that dry FGC processes dominated in the three countries with the highest number of waste incineration plants (France, Germany, and Italy, see Table 3) [71].

**Table 3.** WtE systems with dry or semi-dry flue gas cleaning (FCG) (at least one combustion line), 2011, based on ISWA [71].

| Country | Total WtE Plants | Plants with Data Available | Dry FGC | Semi-Dry FCG | Dry + SD FCG | Wet FGC + Dry or + SD | Sum Dry + SD | Share of Plants with Data Available |
|---|---|---|---|---|---|---|---|---|
| France | 127 | 124 | 47 | 24 | 2 | 2 | 75 | 60% |
| Germany | 55 | 47 | 6 | 19 | 0 | 6 | 31 | 66% |
| Italy | 51 | 49 | 22 | 10 | 1 | 3 | 36 | 73% |

For these plants, an extensive conversion of the flue gas cleaning system would be required. In Switzerland, the Netherlands, Austria, and Sweden, wet or so-called hybrid processes—the combination of a wet and conditioned dry process—dominated. In these countries, however, there was no requirement for wastewater-free plant operation, meaning that the wastewater can be discharged into the receiving water after wastewater treatment. This plant operation naturally has certain economic advantages in terms of residual waste disposal costs [72].

Apart from that, there is another problem: "The municipal waste-to-energy plants do not generate enough raw acid by treatment of household wastes, and the supply gap can be covered either by technical grade hydrochloric acid bought on the market or by in situ generated hydrochloric acid from mechanically non-recyclable PVC wastes" [46]. But a significant increase in PVC throughput volumes poses problems for many of the plants currently in operation, like, e.g., chlorine corrosion on the evaporator heating surfaces by hydrogen chloride. "The operator of plant must ensure that the HCl is separated during waste gas purification to such an extent that the limit values specified in the plant licence are complied with. On the one hand, he must not exceed the maximum HCl concentration in the raw gas before it enters the waste gas purification system, which is the basis for the design of the waste gas purification system, in order not to "overrun" the system. On the other hand, he naturally endeavours to minimise the use of neutralising agents for cost reasons. For this reason alone, he keeps the HCl raw gas concentration within the specified limits by carefully mixing the waste in the bunker" [73]. As early as the 1990s, it was criticised that an increasing proportion of PVC in the MSWI input would result in a higher demand for neutralisation agents [9]. For this reason, PVC is also especially regulated in the acceptance conditions of some waste incineration plants (e.g., exclusion from acceptance or delivery of PVC waste only after prior agreement [74–76]).

A regulation requiring the conversion of the flue gas cleaning of all waste incineration plants in a state to such technologies is politically feasible, as the example of Switzerland shows. But it would be politically and economically challenging for Europe (2020: 504 [77]), as many incineration plants are equipped with dry or semi-dry flue gas cleaning [71]. Notwithstanding, metal recovery from MSWI filter ash is a good idea, also because some of these metals contained in the filter ash are strategically important for the EU (copper, nickel) or even critical (antimony) [78]. In PVC, antimony is used in form of antimony trioxide (ATO, antimony(III) oxide, $Sb_2O_3$) as a flame-retardant [79]. Antimony, furthermore, is not recovered during the FLUWA process, but remains in the fly ash. The implementation of a sequential extraction with different acids (HCl followed by phosphoric acid followed by citric acid) showed high potential for the future to achieve high antimony recovery rates [80], but this is currently not feasible in incineration practice.

## 4. Discussion: No Sustainability—Attempts of an Explanation

### 4.1. Starting Point

More than 30 years have passed since the introduction of the "Plastics Cycle", "PVC cycle guarantee", and "Global Recycling" concepts. However, the post-consumer material recycling rate, as documented above, is meagre. And specific information on the extent of the closed loop respective to the replacement of virgin PVC is not available.

The balance of chemical PVC recycling is currently even worse than that of material recycling. In 2021, around 400 tons of PVC waste (400 t/658,000 t following Conversio ([23], pp. 60–63)) were sent to chemical recycling (presumably use of the mixed plastics in the blast furnace process) in Germany, which is 0.06 percent. In Europe, the figures are likely to be even lower. This result is difficult to understand, as chemical recycling promises good results, at least in chemical theory (virgin plastic) [60,81]. The ecological analysis also shows that—depending on the boundary conditions—chemical recycling of PVC can be advantageous [82,83].

Can this result for Germany (or Europe) be explained by a lack of money or a lack of state support? It was certainly also due to money. However, the call for the state leads to a dilemma: the chemical industry—like all other sectors—actually wants less, not more governmental involvement. And the recycling promise for PVC was positioned from the outset as a voluntary activity by industry. So, what could the reasons be for so little progress?

### 4.2. First Reason: The Quantity and Variety of Additives

PVC is chemically very complex. PVC itself has, as described above, properties that do not make it a favourable polymer, with the key weakness being chlorine. From 100 °C to 120 °C, the polymer begins to decompose and hydrogen chloride (HCl) is released (de-hydrochlorination [84]). This decomposition, in turn, leads to chain reactions. As temperatures in the range of 160 to 200 °C are required for, e.g., extruding, this polymer could not be processed at all without considerable damage. The polymer would therefore simply be unusable as a plastic, especially for a second or third processing at elevated temperatures in the context of recycling. This is where additives (stabilizers) come into play [84]. These are added to the polymer before the first "melting" (in the sense of "making malleable"). Although they cannot completely prevent HCl decomposition, the additives do stop the chain reaction.

Solar radiation (heating, UV) also leads to the described de-hydrochlorination and chain reaction [84]. This also causes the polymer to age quickly, and should be considered unsuitable for most applications. Stabilisers—mostly metal soaps, metal salts (calcium/zinc), or organometallic compounds (organotin)—are usually combined with organic co-stabilisers (polyols or epoxidized esters) [84]. The stabilisers are used up in full or in part during the lifetime of a product (through the protective reaction). Strictly speaking, therefore, the stabilisation status would have to be determined by chemical analysis before re-processing. This is an enormous amount of work that cannot be carried

out in practice by small- and medium-sized companies. You can expect larger companies to be able to make this effort.

Another disadvantage of the PVC polymer poses the intra-molecular forces caused by the chlorine (dipole forces and size of the chlorine atom). The polymer is brittle and stiff, and is therefore unusable. It has to be made flexible. Other additives (plasticisers) are used for this purpose. Over 85% of the plasticisers used in plastics in Europe end up in PVC [21]. The weaknesses of the polymer therefore mean that PVC is one of the plastics with the highest and most complex additive content, which would make high-quality recycling of mixed post-consumer waste impossible. There are 470 different PVC additives in use today [21]. Therefore, even if the waste is collected by product type, the diversity is still so high that often only downcycling or more often disposal (today mostly incineration) are the only options.

It can also be assumed that the plastic is damaged during the second or further melting during the recycling process (high temperature, shear forces, presence of oxygen, carbonyl, and peroxide compounds). Processing therefore leads to changes in the polymer molecule, which in turn causes an increase in the mobility of the additive molecules ([85], cited by [86]). We therefore do not believe in the narrative of infinite cycles (circular economy), especially for PVC. 'Vinyl', in contrast, refers to experiments that have shown that PVC can withstand 10 thermal cycles without addition of further stabilisers or process aids. In [87], the experiments were performed with two PVC-U (rigid-PVC) suspension resins, the one containing a tin stabilizer (2%), the other containing a lead stabilizer (8%), plus further additives (e.g., pigmentation). Both additives are meanwhile banned (see Section 4.3). "The study has confirmed that the physical properties of recycled PVC-U are very similar to virgin material even after several passes". It should be noted that this was a material that was not subject to any further changes in the course of the investigation and was not mixed with other PVC materials that were additivated differently. The authors themselves limit the significance of their test results with regard to the first point: "The process did not allow normal ageing between the cycles which according to some researchers may be expected to somewhat reduce the thermal stability of the material (. . .). . . . Clearly such modifications could be utilised in practice whereby the recycled material may not necessarily be 100% recycled but blended with virgin material". The other study [88] cited by 'Vinyl', using one dry blend (pipe formulation) stabilized with Ca/Zn (2.8%), confirmed the findings of the first [87].

It therefore remains an open question, whether we can reach in reality one cycle, two cycles, or more with PVC. But this is not "circular economy".

### 4.3. Second Reason: Legacy Chemicals (Legacy Additives)

There is another chemical reason, which is related to the first reason and which makes it even more difficult for the processing industry (converters) to achieve a good quality of the recyclates. Particularly for toxicological reasons, many of the additives previously used as stabilisers and plasticisers in PVC products have since been banned. Due to the long product lifetimes of some PVC articles, they are still in circulation [89]. PVC recycling today leads to products that still contain these banned additives through the recyclates in them [53,90]. The long list of banned PVC additives [91] includes phthalates, short-chain chlorinated paraffins, hexabromocyclododecane (HBCDD), organotin compounds, arsenic, lead, cadmium, nickel, and chromium.

The group of phthalates alone comprises around a dozen individual substances, and the above list of regulated additives is not final [92]. And the discussion about further bans is not over, on the contrary [21]. For some additives that are banned, there are increased limit values for substance concentrations in recyclates (e.g., DEHP, Cd, Pb) or other requirements to not hamper too much the efforts towards recycling targets. Therefore, it is very difficult for recyclers and converters to maintain an overview, let alone determine what banned pollutants (and candidate substances [93]) are contained in the PVC waste they receive. Also, a glance at the EU database SCIP (Substances of Concern In Products),

which was put in place to better inform actors at the end-of-life treatment stage about harmful chemicals in waste, cannot solve this problem [94]. A solution for ending this vicious circle needs to be found.

'Vinyl' states: "However, it is true that the growth rate of recycling has slowed down in the last decade . . .. In some cases, uncertainty regarding upcoming restrictions of legacy additives, and whether recycling would still be possible (with appropriate limit values) has led to postponed investment. This is the case for the restriction on lead in PVC, which was adopted last year and now foresees a 10-year derogation for recycling lead-containing PVC that meets specific conditions [95]. Future restrictions, such as on MCCPs [96], also have an impact. VinylPlus will consider those impacts in view of the Voluntary Commitment target review in 2025". However, this argument for the decline in recycling must also include the argument that the ban on an additive was preceded by the decision to have used this additive at all without sufficient safety tests.

### 4.4. Third Reason: Economic Interests?

Apparently small and medium-sized plastics processors (converters) have no economic problem with an increasing use of recyclates, if (simplified) the recyclates are of sufficient quality and at good prices. One step before in the supply chain, the situation is different for PVC manufacturers (especially for polymer producers). Even a temporary drop in demand triggers concern [97]: "The European—and in particular the German—PVC market is staggering at the brink of destruction. At least that is what a look at the current production figures and the results of a recent survey of the most important PVC producers in Europe suggest" ("Der europäische—und insbesondere der deutsche—PVC-Markt taumelt am Rand des Abgrunds. Das jedenfalls legen der Blick auf die aktuellen Produktionszahlen sowie die Ergebnisse einer aktuellen Umfrage unter den wichtigsten PVC-Erzeugern in Europa nahe."). A permanent reduction in the sale of virgin PVC through the expansion of recycling would significantly reduce plant capacity utilisation, which would lead to significant financial losses for PVC manufacturers. It is unlikely that PVC manufacturers will voluntarily bring about and finance this development.

Today, it is no longer disputed that PVC has a central function in chlorine removal. A PVC ban would lead to a chlorine surplus in the chemical industry, which would then become a waste problem. The Vinyl Environmental Council (VEC), for example, argues this to the European Chemicals Agency [21].

But how do these serious negative economic consequences of a ban fit in with the communication of "Vinyl", which propagates the circular economy, which would have the same consequences as a ban?

Unfortunately, we were not able to find a complete data set on the production of PVC polymers or PVC plastics (compounds) at European level. But for single years, data are available. According to PlasticsEurope [98] (there: page 16), the demand for (not: "production of") PVC plastics was around 4.9 million tonnes in 2012. Using the ECHA conversion factor for PVC polymers to PVC compounds in [21], the 4.9 million tonnes of PVC plastics should correspond to around 4 million tonnes of PVC polymer.

Ciacci et al. [99], who were allowed to evaluate the European PVC statistics of PlasticsEurope almost ten years ago, also put the flow of PVC polymers in Europe in 2012 at 4 million tonnes. So, we have two sources for 2012. For a rough estimate of the development, the data for demand for, and the flow of PVC polymers in the EU in 2012 should be equated with sales volumes. The sales volume of PVC polymers in 2021 amounted to 5.2 million tonnes (Eurostat, 2023b, cited in [19]). This means that the increase in sales of PVC polymers over ten years (2012–2021) totalled (5.2 minus 4.0) 1.2 million tonnes, an average increase of 3% or 135,000 tonnes of polymer per year. For PVC compounds, the corresponding sales volumes are 4.9 million tonnes for 2012 and 6.8 million tonnes for 2021 [21], which corresponds to an average increase of 3.7% or 210,000 tonnes of PVC compounds per year.

In the same period (2012–2021), the increase in recyclate volumes from PVC compounds in Europe averaged only around 50,000 tonnes per year (pre- and post-consumer) according to Figure 4; this corresponds to around a quarter of the average growth in production. The quantities of recyclates are even lower if the quantities from post-consumer recycling (at calculation point (2)) are taken into account (less than 4000 tonnes a year). And the volume of tonnes that have replaced virgin PVC (calculation point (4)) is surely even lower. But only the latter would affect the PVC industry's sales.

The production of PVC polymers (or PVC compounds) in the EU has de facto increased far more than the quantities of substitution of virgin PVC from recycled waste PVC in the period 2012–2021.

## 5. Conclusions

The PVC industry has been promising a sustainable PVC cycle for over 30 years. The current level of certified material recycling of post-consumer waste in Europe is a meagre 12%. Vinyl has also informed us that post-consumer recycling volumes will amount to 599,000 tons in 2020. However, these figures are extrapolated and not certified. And we do not know if there is a substantial reduction of virgin PVC with recyclates. And some of the resulting products are of low quality and are frequently contaminated with banned pollutants (legacy additives).

'Vinyl's' current annual reports show significantly higher figures (see above), mainly because pre-consumer recyclates are also included. The EU Commission has allowed this practice exclusively for the PVC industry so that it can fulfil its voluntary commitments. However, the main contributions to closing a material cycle can only be made through post-consumer recycling. We therefore propose that only post-consumer recyclates should be counted for successful material recycling.

As early as 1995, experts came to the following conclusion regarding the recycling of PVC [100]:

"For individual product groups, a real material and product cycle would have to be established by the manufacturers concerned, leading to a reduction in new production in line with the quantities of recyclate produced. This would require a new industrial policy in the respective sectors. New production and recyclate production should no longer take place separately in terms of organization and business management, but only in a fully integrated process". ("Für einzelne Produktgruppen müsste von den jeweils betroffenen Herstellern ein wirklicher Stoff- und Produktkreislauf aufgebaut werden, der zu einer entsprechend den erzeugten Rezyklatmengen verringerten Neuproduktion führt. Dies würde eine neue Industriepolitik in den jeweiligen Branchen voraussetzen. Neuproduktion und Rezyklatproduktion dürften organisatorisch und betriebswirtschaftlich nicht mehr getrennt ablaufen, sondern nur in einem vollintegrierten Prozess.").

On the road to a circular economy, the market for virgin PVC was supposed to gradually shrink in order to be sustainable. This has not happened. The sale of PVC (compounds) in Europe has risen steadily—with dips since the 1990s—by around 210,000 tons per year in the last ten years alone. The annual increase in PVC recyclates (total post- and pre-consumer) over the same period was only around 50,000 tons per year (see Section 4.4), and the increase in the post-consumer recycling alone was not even 4000 tons per year. The substitution of virgin PVC can only be a share of these 4000 tons. We therefore also suggest that the substitution of virgin PVC with post-consumer recyclates should be collected statistically.

At the beginning of the 2000s, optimism still prevailed with regard to chemical recycling: "It is assumed that, by the end of 2002, most of this information will be made available to form a basis from which industry can select and take investment decisions" [101]. But in the last 30 years, not a single industrial plant for chemical recycling has been built and successfully operated. And there has also been no practical recycling solution for chlorine, which accounts for 57% of pure PVC polymer by weight. Despite the promise of recycling,

the lion's share of PVC waste in Europe is still being incinerated in waste-to-energy-plants, where it tends to be a nuisance [75–77].

At the end of 2023, the European Chemicals Agency ECHA presented an "Investigation Report" following a request of the EU Commission [21]. In the second part, we will therefore analyse what regulatory consequences need to be drawn.

**Author Contributions:** Conceptualization, writing—original draft preparation, U.L.; writing—review and editing, B.Z.-L. All authors have read and agreed to the published version of the manuscript.

**Funding:** This research received no external funding.

**Institutional Review Board Statement:** Not applicable.

**Informed Consent Statement:** Not applicable.

**Data Availability Statement:** The original contributions presented in the study are included in the article, further inquiries can be directed to the corresponding author.

**Acknowledgments:** The authors would like to sincerely thank Christine Herrmann (European Environmental Bureau (EEB)), Klaus Günter Steinhäuser (Arbeitskreis Umweltchemikalien/Toxikologie beim Bund für Umwelt und Naturschutz Deutschland e.V. (BUND)), Vincent Stone (European Council of Vinyl Manufacturers/VinylPlus) and Geoffroy Tillieux (European Plastics Converters (EuPC)) for their valuable feedback.

**Conflicts of Interest:** Authors Uwe Lahl and Barbara Zeschmar-Lahl were employed by the company BZL Kommunikation und Projektsteuerung GmbH. The remaining authors declare that the research was conducted in the absence of any commercial or financial relationships that could be construed as a potential conflict of interest.

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
