# Peer review of "More than 30 Years of PVC Recycling in Europe—A Critical Inventory"

_sustainability, doi:10.3390/su16093854_

Round 1

Reviewer 1 Report

Comments and Suggestions for Authors

This paper analyzes the significance of replacing virgin PVC with recycled material, and from this it is concluded that after a good thirty years, the results of recycling are quite meager. In general, I would suggest a major revision of this paper upon addressing the following concerns:

1.      Please indicate on what basis the article speculates that virgin PVC can be completely replaced with recyclables by 2040.

2.      In figure 5, please analyze the reasons for the fluctuating development curve of global PVC production and why it plummeted between 2008 and 2009.

3.      What is the innovation of this article and how does the material compare to previous research.

Author Response

Dear reviewer!

Thanks for your efforts. Please see the attachment.

Kind regards

Barbara Zeschmar-Lahl

Reviewer 2 Report

Comments and Suggestions for Authors

The presented article is well-written and organized discussing a sustanability-related PVC recycling issue. The review will serve as a good reference for those who are interested in PVC recycling history or for researchers who are involved in chemical recycling and sustainability. The paper can be accepted after following major revision being addressed:

1-In the introduction part, authors are advised to show the scheme for polymerization of VC monomer to PVC. Also, they need to show why the PVC is highly toxic among other polyolefins such as PE, PP and PS. Provide types of PVC and their main compositions/applications. Also, mention briefly the main characteristics of PVC such as low thermal stability, degradation at low temperature, toxicity. This will be helpful for later discussion in the review. 

2- Likewise, authors need to show in the introduction the main applications of the PVC industry and what industries are banned from using PVC.

3- In page 5, what does rPVC means. Pls. define all acronyms/abbreviations at the first time of its appearance inside the text. 

4- section 2.3 is written in a complicated form, pls. try to brief/sumerize the information about recycling with a logical flow. You may use scheme/diagram to illustrate the types of recycling. You may mention mechanical recycling, primary, secondary and tertiary recycling if there is a chance to correlate with PVC. Also, you need to  introduce chemical/feedstock recycling which is discussed later in the review. 

5- it is stated in page 6 that "It is clear that post-consumer recycling results are mainly achieved through the recycling of window frames (56%)". Is there reasons to clarify as compared to very low recycling of other types pf PVC products (e.g. flooring)? This may lead to understanding that some products has to be banned as they are difficult to be recycled.

6- It would be interesting if there would be some information about the current burning practices of PVC and its contribtion to air pollution.

7- It is suggested to provide the recycling levels of PVC worldwide. The main discussion is in Europe and Germany. This could provide support for the current status of PVC recycling. 

8- Authors did not discuss the substitution of PVC with other polyolefins or other thermoplastics. Why it was not suggested? Is it feasible or NOT? This could be a real solution to replace PVC either soft, rigid or both.

Author Response

(The authors gave the same response as above.)

Reviewer 3 Report

Comments and Suggestions for Authors

The paper “sustainability-2988495” is focused on statistics, plans and promises vs. reality in the field of industrial PVC recycling in Western Europe. The paper raises an important problem and gives quite dire outlook for the PVC recycling field. In my opinion the authors should revise the paper according to the following points.

1. Formatting of the paper seems not to follow MDPI style.

2. There are a lot of footnotes in the paper text. It makes the text more hard to read. In some places footnotes can be replaced by references and some footnotes can be moved to the main text.

3. There is no need to write “own graphic” in the caption for the figures.

4. There are a lot of abbreviations that, although quote well-known (such as DDT, DEHP etc.) should be spelled out at first appearance in the text.

5. In lines 40-44 the authors give references for the first two points of the bullet list, but not for the third one.

6. In line 86 the authors probably misprinted 15 as “35”.

7. The answer to the question in Lines 151-153 seems quite obvious. Of course, after a thorough (and mixing should be thorough) mixing of virgin PVC to the recycled PVC their separation is impossible in practice. This is not a question, this is a problem for the recycling of products obtained from already recycled material.

8. In my opinion, there is a contradiction between some statements in the paper text.

In footnote 11 it is stated that “post-consumer recycling <increased> from 2000 until 2020 from 100 kT to 599kT”. The authors do not argue with this statement despite the doubts of its significance.

At the same time in Line606 the authors wrote that “the increase in the post-consumer recycling alone was not even 4,000 tons per year”. However from footnote 11 it follows that mean increase was at least (599-100)/20 ~ 25 000 tons per year.

9. In Line 570 the authors estimate the production of PVC to about 6.8 million tonnes per year. Taking into account data from Table 1 and Lines 564-573, a rough estimation of PVC production in 2000 of 3.5 million tonnes can be made. Thus, again taking into account the data from footnote 11, it can be calculated that a fraction of recycled PVC in 2000 was lower than 3% and by the 2020 it rose up to almost 9%. Is this percentage still low? Probably so. Should we try to increase it? Yes, again. But are we moving in the right direction? From the numbers it seems that we are. I agree that a high annual growth percentage when the base value is very low is not always something to be proud of. But in this particular case I would not be so harsh to claim that the fact of increase of the post-consumer recycling from 2000 until 2020 from 100 kT to 599kT is of little significance, as the authors were.

10. For me, as a researcher in the field of polymer science, appeals such as in Lines 265-266 seem to be out of place for the scientific article. However if that is considered ok in this field, this remark can be dismissed.

11. In lines 296-302 it is noted that due to presence of hazardous substances, post-consumer material may only be used for the internal profiles. My question is, if the internal profile is in direct contact with the external profile and the hazardous substances can still migrate to the surface and even in air via diffusion, how is such problem dealt with?

12. In line 335 and in the box in page 19 there is a missing footnote

13. Line 452. For what application antimony is added to the PVC materials?

14. Line 477. Plasticization is the softening, increase of plasticity, flexibility, decrease of viscosity due to addition of low molecular compound (plasticizer) to the polymer. Softening of the polymer due to temperature increase is not a plasticization.

15. Line 498. From strict thermodynamic point of view melting of PVC is very rare phenomenon since it is almost completely amorphous. It is true that for processing of PVC increased temperatures are often used but no melting occurs in these processes since there is nothing to be melted in the PVC (although 1-2% of crystallinity degree is sometimes noticed).

16. Footnotes 29,30 are in Deutsch, please translate it to English.

17. In footnote 31 2022 is probably misprinted as “2002”.

18. Since the paper focuses on the state of the problem in European union, in Germany especially, with almost no mention of other countries, it should either be noted in the title of the paper, or else the authors should describe other regions as well.

Author Response

(The authors gave the same response as above.)

Round 2

Reviewer 1 Report

Comments and Suggestions for Authors

The authors have well replied to the raised questions, and the obtained data have well supported the conclusions, so it is recommended to accept this paper.

Reviewer 2 Report

Comments and Suggestions for Authors

Authors have responded positively to all comments and suggestions. They have provided detailed explanation and supported their revision with schemes and up to date references. The manuscript can be accepted at this stage.

Reviewer 3 Report

Comments and Suggestions for Authors

In my opinion, the revised version of the paper can be published. In the proof stage of publication missing references in lines 98 and 117 should be added.